# Structural and biochemical analysis of ligand binding in yeast Niemann–Pick type C1–related protein

Lynette Nel[1,*], Katja Thaysen[2,*], Denisa Jamecna[3,*], Esben Olesen[1], Maria Szomek[2], Julia Langer[3], Kelly M Frain[1], Doris Höglinger[3], Daniel Wüstner[2], Bjørn P Pedersen[1]

In eukaryotes, integration of sterols into the vacuolar/lysosomal membrane is critically dependent on the Niemann–Pick type C (NPC) system. The system consists of an integral membrane protein, called NCR1 in yeast, and NPC2, a luminal soluble protein that transfers sterols to the N-terminal domain (NTD) of NCR1 before membrane integration. Both proteins have been implicated in sterol homeostasis of yeast and humans. Here, we investigate sterol and lipid binding of the NCR1/NPC2 transport system and determine crystal structures of the sterol binding NTD. The NTD binds both ergosterol and cholesterol, with nearly identical conformations of the binding pocket. Apart from sterols, the NTD can also bind fluorescent analogs of phosphatidylinositol, phosphatidylcholine, and phosphatidylserine, as well as sphingosine and ceramide. We confirm the multi-lipid scope of the NCR1/NPC2 system using photo-crosslinkable and clickable lipid analogs, namely, pac-cholesterol, pac-sphingosine, and pac-ceramide. Finally, we reconstitute the transfer of pac-sphingosine from NPC2 to the NTD in vitro. Collectively, our results support that the yeast NPC system can work as versatile machinery for vacuolar homeostasis of structurally diverse lipids, besides ergosterol.

## Introduction

Sterols are indispensable components of membranes in all eukaryotic cells and precursors to many metabolites, which are required for various cellular processes (1, 2, 3). Endogenous sterols are synthesized at the ER, and exogenous sterols can enter the cell through, for example, endocytosis (4). The redistribution of sterols to all membranes of the cell occurs, for instance, at acidic organelles called lysosomes in animals and vacuoles in fungi and plants (5, 6, 7, 8, 9, 10, 11). The luminal membrane of lysosomes and vacuoles is lined with a glycocalyx, a polysaccharide matrix that prevents autodigestion by resident hydrolytic enzymes (12, 13, 14, 15, 16).

In humans, the Niemann–Pick type C (NPC) disease is a lysosomal storage disorder characterized by intralysosomal accumulation (storage) of cholesterol and sphingolipids, typically manifesting as progressive neurological dysfunction in children (17, 18). The disease arises from mutations in genes encoding either the Niemann–Pick type C1 protein (hNPC1, an integral membrane protein) that integrates sterols into the lysosomal membrane, or the Niemann–Pick type C2 protein (hNPC2, a soluble protein inside the lysosomal lumen) that delivers sterols to hNPC1 (19, 20, 21, 22). Homologs of both proteins are found in all eukaryotes. The vacuole of *Saccharomyces cerevisiae* contains Niemann–Pick type C1–related protein 1 (NCR1) and NPC2. Notably, NCR1 and NPC2 can rescue the disease phenotype in NPC-deficient mammalian cells, making it a suitable model system to better understand NPC disease and eukaryotic sterol and lipid homeostasis in general (23, 24).

NPC2 proteins consist of seven anti-parallel $\beta$-sheets that form a hydrophobic sterol binding pocket, with NPC2 having a ~fivefold larger binding pocket compared with that of human and bovine NPC2 (25, 26, 27, 28). NCR1 and hNPC1 share the same structural fold: an N-terminal domain (NTD), a middle-luminal domain, and a C-terminal domain found on the luminal side of the vacuolar/lysosomal membrane. Thirteen transmembrane helices (M1–M13) span the membrane, of which M1 anchors the NTD to the vacuolar/lysosomal membrane, whereas the remaining 12 transmembrane helices are divided into a sterol-sensing domain (SSD, M2–M7) and a pseudo-SSD domain (pSSD, M8–M13) (14, 22, 27, 29, 30). This fold is characteristic of the resistance–nodulation–division superfamily, but with the NTD and M1 helix being unique features found only in NPC proteins (31, 32). The current transport model for the NPC system postulates that NPC2 binds sterols in the lumen and loads them into the NTD, which transfers the sterol to a tunnel that is formed by the middle-luminal domain and C-terminal domain of hNPC1 and NCR1, respectively (1, 27, 33, 34). This tunnel, essential to bypassing the glycocalyx, changes shape depending on the protonation state of key residues in the SSD and pSSD, to transport the

[1]Department of Molecular Biology and Genetics, Aarhus University, Aarhus, Denmark   [2]Department of Biochemistry and Molecular Biology, University of Southern Denmark, Odense, Denmark   [3]Heidelberg University, Biochemistry Center, Ruprecht-Karls-Universität Heidelberg, Heidelberg, Germany

Correspondence: wuestner@bmb.sdu.dk; bpp@mbg.au.dk
*Lynette Nel, Katja Thaysen, and Denisa Jamecna contributed equally to this work

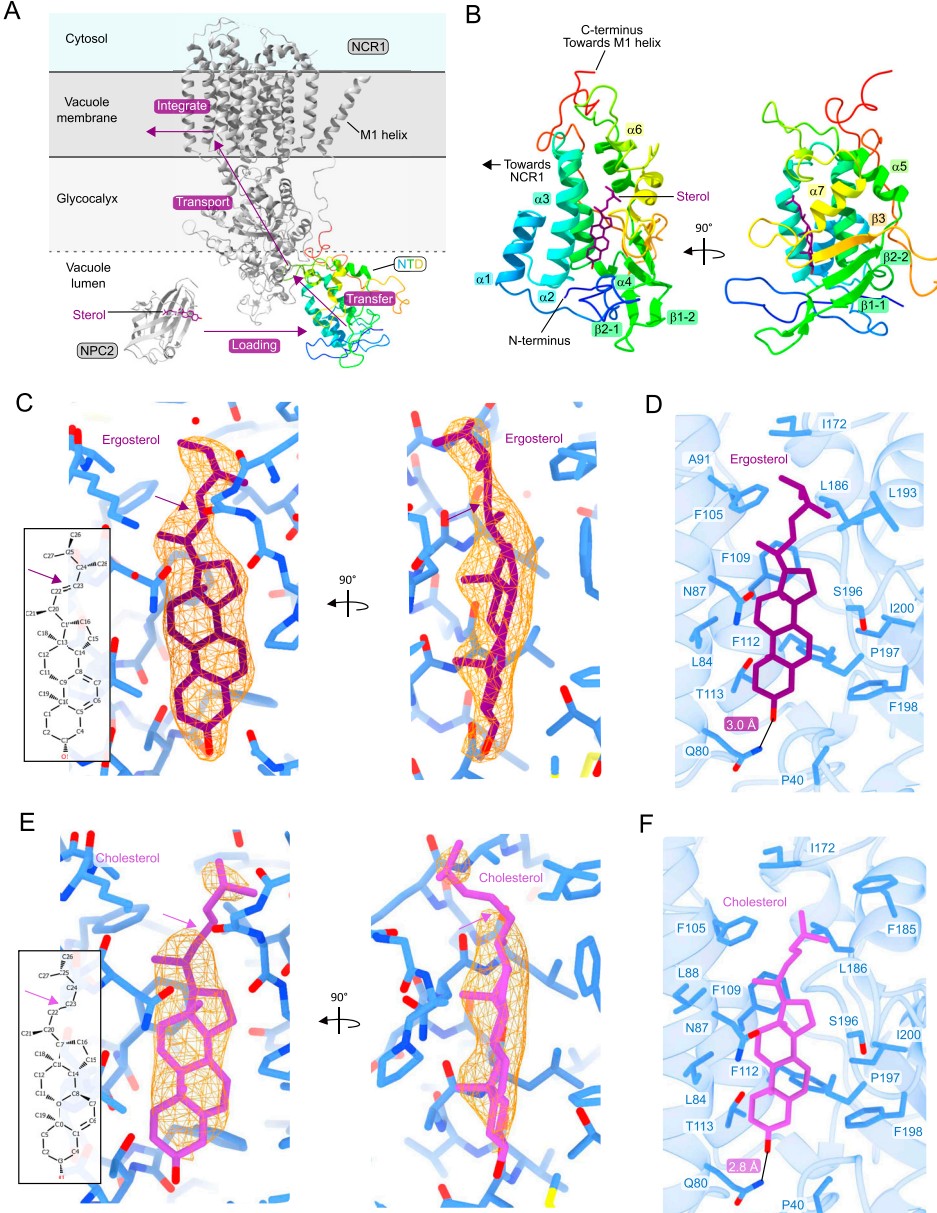

**Figure 1. Structures of the NTD bound to sterols.**
**(A)** Structure of NCR1 (PDB ID: 6R4L) in gray with the NTD in color. NTD loading from NPC2 (PDB ID: 6R4N) is followed by transfer, transport, and integration by NCR1 into the vacuole membrane. **(B)** Secondary structure elements of the NTD. The NTD has seven α-helices, which are interrupted by two β-sheets between α4 and α5. After α7, the third β-sheet connects the NTD with the long loop leading to the M1 transmembrane helix. The color gradient starts as blue at the N-terminus and transitions to red at the C-terminus. **(C)** Chemical structure and density of ergosterol in the binding pocket of the NTD. The double bond between C22 and C23 makes the aliphatic tail of ergosterol rigid and can be seen within the continuous density in orange surrounding the molecule. **(D)** Residues surrounding ergosterol in the binding pocket of the NTD are mostly hydrophobic, except for Q80, N87, T113, and S196. The hydroxyl group of ergosterol is coordinated by Q80, 3.0 Å away. **(E)** Chemical structure and density of cholesterol in the binding pocket of the NTD. In cholesterol, the double bond is lacking between C22 and C23 and makes the aliphatic tail more flexible, as can be seen in the surrounding discontinuous orange density. **(F)** Residues surrounding cholesterol in the binding pocket of the NTD are the same as for ergosterol. The hydroxyl group of cholesterol is closer to Q80, 2.8 Å away.

sterol to the luminal membrane for membrane integration (Fig 1A) (14, 22).

It has been shown that the NTD of hNPC1 binds cholesterol and 25-hydroxycholesterol tightly (35). The binding pocket of hNPC2 can accommodate a variety of sterols including cholesterol, epi-cholesterol, and cholesteryl sulfate, but its affinity for oxysterols is much lower (36, 37, 38). Synthesis and transport of cholesterol derivatives, such as 25- and 27-hydroxycholesterol, are disturbed when cells lack hNPC2, but this protein is not strictly needed to transport fluorescent analogs of 25- and 27-hydroxycholesterol out of the lysosomes in human fibroblasts from NPC disease patients (36, 38, 39). In addition to sterols, hNPC2 interacts with various phospholipids, of which the highest relative binding was observed for lysobisphosphatidic acid, a lysosomal-specific phospholipid

(40). It was previously shown that the large binding pocket of yeast NPC2 enables it to bind a variety of substrates in vitro, such as edelfosine (a lipid analog with antifungal properties) (41, 42), U18666A (a cationic sterol that inhibits Ebola infection through NPC1) (43), sterols, phospholipids, and sphingolipids (27, 28). The broad substrate binding capacity of yeast NPC2, when compared to the human counterpart, and its capability to transfer sterols to the NTD of NCR1 led to the speculation that the NTD might bind a diverse range of lipids, and not only sterols (27, 28).

In this study, we examine the substrate specificity of the NTD of NCR1 and NPC2. We describe structures of the NTD bound to either er-gosterol or cholesterol and analyze the architecture of the substrate binding pocket. In vitro binding of ligand candidates to delipidated NTD and NPC2 is evaluated using nitrobenzoxadiazole (NBD)-tagged

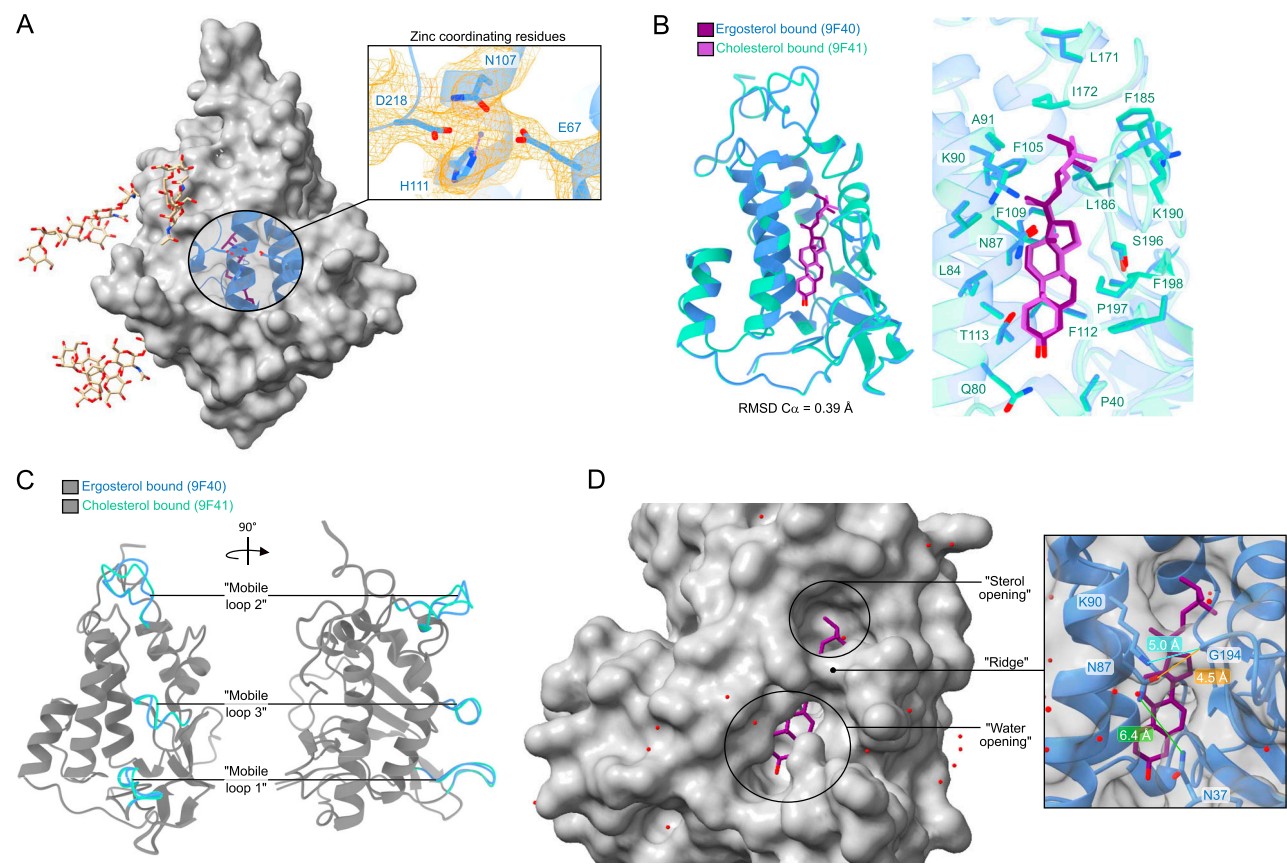

**Figure 2. Structural features of the NTD.**
**(A)** Ion-coordinating residues are found at the opposite face of the substrate binding pocket. Density surrounding residues E67, N107, H111, D218, and the zinc ion are shown in orange. **(B)** Superposition of NTD structures bound to ergosterol and cholesterol. The sterols and residues of both binding pockets are overlain and show the same positioning between the two structures. **(C)** When overlaying the NTD bound to ergosterol and cholesterol, three "mobile loops" are displaced when comparing the two structures. These mobile loops cover the substrate binding pocket at the bottom, middle, and top. **(D)** Residues that form the "ridge" include N87, K90, and G194—with G194 being on "mobile loop 3"—and likely govern substrate accessibility to the binding pocket. The "ridge" divides the pocket into a "sterol opening," with the aliphatic tail of the sterol being visible, and a "water opening," which houses the hydroxyl group of the sterol.

lipids and photo-activatable and clickable (pac-) lipids (28, 44, 45, 46). The results indicate that the NTD and NPC2 bind pac-modified cholesterol, and NBD- and pac-phospholipids and sphingolipids. Lastly, we demonstrate that NPC2 can transfer pac-sphingosine to the NTD. Overall, these results expand our knowledge regarding the capacity of NPC2 and NCR1 to bind a range of hydrophobic ligands.

# Results

### Ergosterol- and cholesterol-bound structures of the NTD

To study the substrate scope of NCR1, the NTD (residues 1–249 of NCR1) was purified and crystallized with either ergosterol or cholesterol (Fig S1A, Table S1): ergosterol-bound structure at 2.4 Å ($R_{free}$ 26%) and cholesterol-bound structure at 2.6 Å ($R_{free}$ 26%) (Fig 1B–F). Besides the sterol, the two structures show minimal differences between them (root mean square deviation of C-alpha atoms,

$RMSD_{C\alpha}$, of 0.39 Å). In the asymmetric unit, there are four monomers of the NTD, with no structural differences between monomers ($RMSD_{C\alpha}$ of 0.13 to 0.26 Å).

The architecture of the NTD consists of seven $\alpha$-helices and a three-strand mixed $\beta$-sheet (Fig 1B). All 16 cysteine residues form disulfide bridges and are partitioned to the side facing the other luminal domains of NCR1 (Fig S1B). Three residues (N123, N145, and N178) are *N*-glycosylated, and deglycosylation of the purified NTD yielded a sample that was reduced by ~8 kD (equivalent to about 44 mannose units) (Fig S1C). Both structures were derived from glycosylated samples and show long glycosylation chains that extend from the "back" of the NTD, the opposing side to the substrate binding pocket (Fig S1B and D). The glycan chains are well structured, and three to seven sugars can be modeled per glycosylation site, in some to the point where the glycan chain starts to branch (Fig S1E).

The clear density of an ion in a well-defined cation coordination site on the "back" of the NTD was identified (Fig 2A). Because ZnSO₄ was used in the crystallization solution, a zinc ion was modeled. The site is formed by E67 (on the loop connecting helices 1 and 2), N107,

H111 (both on helix 3), and D218 (on the loop that becomes the linker toward M1).

Ergosterol and cholesterol are highly similar in their chemical structure (Fig 1C and E), but the quality of the electron density maps allows for clear discernment between the two in the NTD binding pocket. In particular, the double bond between C22 and C23 of ergosterol results in a more rigid iso-octyl chain, evident as density that is continuous with the ring system (Fig 1C). Cholesterol lacks this double bond in the iso-octyl chain, making it more flexible, and this is observed as density that is discontinuous with the ring system (Fig 1E). The substrate binding pockets of the two structures are virtually identical, with both substrates positioned parallel to helix 2 (Fig 2B). The interior of the substrate binding pocket is lined with hydrophobic residues, except for Q80, N87, T113, and S196—with Q80 coordinating the hydroxyl group of the sterol (Fig 1D and F). F109 and F112 are positioned underneath the ligand, allowing for pi-stacking with the sterol ring system. Minor differences between the two structures are observed in three mobile loops: "mobile loop 1" leads to helix 1, "mobile loop 2" connects helix 5 to helix 6, and "mobile loop 3" connects helix 7 to strand 3 (Figs 1B and 2C). Upon comparison, the position of "mobile loop 2" is displaced the most when aligning the ergosterol- and cholesterol-bound structures.

The binding pocket of the NTD is closed off by a "ridge," formed by the sidechain of K90 and the backbone carboxyl group of G194 (5.0 Å apart), dividing the binding pocket into a "sterol opening" and "water opening" (Fig 2D). In addition, N37 and N87 partially cover the "water opening," with their sidechains ~6 Å apart. In this way, the binding mode of the sterol is with the hydrophilic headgroup the deepest in the pocket, whereas the hydrophobic tail extends toward the sterol opening.

## The NTD and NPC2 bind a range of NBD-tagged lipids

To investigate substrate candidates for the NTD of NCR1 and NPC2, biochemical assays were conducted with purified protein and fluorescent lipid analogs. The purified, delipidated NTD was titrated into NBD-tagged phospholipids and sphingolipids, in which the fluorophore is attached to the hydrophobic tail of the lipid molecules. An increase in signal is based on an enhanced molecular brightness of the NBD-moiety upon binding in a more hydrophobic environment (28). This increase in fluorescence correlates with the concentration of the NTD (Fig 3A), which can be used to infer the affinity of the NTD to a given NBD-lipid (Fig 3B–G). Binding curves of the NTD with five NBD-lipids were obtained: phosphatidylcholine (NBD-PC), phosphatidylinositol (NBD-PI), phosphatidylserine (NBD-PS), ceramide (NBD-Cer), and sphingosine (NBD-Sph) (Fig 3B–G). The measured increase in fluorescence of each NBD-lipid upon titrating the protein was used to determine an apparent dissociation constant ($K_D$) according to a one-site binding model (Equation (1)).

Most measurements were carried out at pH 5.5 to mimic the physiological pH of the vacuole, but the affinity of the NTD to NBD-lipids at neutral pH (7.5) was also assessed. The NTD has the strongest binding affinity for NBD-PI ($K_D$ ≈ 210 nM) and NBD-Sph ($K_D$ ≈ 250 nM), followed by NBD-PC ($K_D$ ≈ 570 nM), NBD-PS ($K_D$ ≈ 930 nM), and NBD-Cer ($K_D$ ≈ 1,260 nM) at pH 5.5 (Table 1). For NBD-PI and NBD-Sph, the concentration used in the binding assays of 1 µM is below

their respective critical micelle concentration (CMC), which was measured to be 2.3 µM for NBD-PI (28) and ~13 µM for NBD-Sph with the same assay (Fig S2A). Thus, for these two lipids, the increase in NBD fluorescence likely comes from a higher quantum yield of the fluorophore inside the binding pocket but not from dequenching, as observed for micelles of NBD-lipids (28). Measurements with NBD-Sph revealed a twofold stronger binding affinity at neutral compared with acidic pH ($K_D$ ≈ 120 nM compared with $K_D$ ≈ 250 nM, Fig 3F and G). This change in the binding affinity of NBD-Sph to the NTD of NCR1 could be caused by an altered hydrogen binding capacity of the sphingosine backbone when the pH changes, as has been shown to take place for natural sphingosine (47). The latter has a pKa of 6.61 in the aggregated form, so an alteration in the protonation state of NBD-Sph at pH 7.5 compared with pH 5.5 could influence the affinity of NBD-Sph to the NTD.

The ability of NPC2 to bind phospholipids has been shown before (28). In the current study, the binding of NBD-Cer and NBD-Sph to purified NPC2 (Fig S3A) was assessed to compare with that of the NTD. At pH 5.5, the binding affinity of NPC2 for NBD-Cer ($K_D$ ≈ 90 nM, Fig S3B) is stronger compared with that of the NTD ($K_D$ ≈ 1,260 nM, Fig 3E). Both NPC2 and the NTD bind NBD-Sph, but the NTD has a higher affinity ($K_D$ ≈ 250 nM at pH 5.5 and $K_D$ ≈ 120 nM at pH 7.5, Fig 3G and F) than NPC2 ($K_D$ ≈ 580 nM at pH 5.5 and $K_D$ ≈ 530 nM at pH 7.5, Fig S3C and D).

Edelfosine is a synthetic lysophospholipid with cytotoxic antifungal effects. Yeast strains with NCR1 knocked out are resistant to this compound, suggesting that edelfosine is a ligand of NCR1 (41). To investigate this, two competitive binding assays with edelfosine and NBD-Sph were designed (Fig 4). For the first assay, the NTD was saturated by NBD-Sph and the addition of edelfosine led to the exchange of the NTD-bound substrate, causing a drop in NBD-Sph fluorescence by about 50% (Fig 4A). The CMC of edelfosine is measured to be 0.715 µM (Fig S2B); thus, after exceeding this concentration of edelfosine, excess NBD-Sph can participate in micelle formation, which is most likely what is observed as an intensity increase after the addition of 1 µM edelfosine to 1 µM NBD-Sph (Fig S2C). Interestingly, the raw intensities of NBD-Sph at maximal concentration of edelfosine are comparable in the presence and absence of the NTD (not shown). This suggests that most of NBD-Sph is efficiently outcompeted and replaced by edelfosine in the binding pocket of the NTD. To further evaluate the capability of NBD-Sph to participate in micelle formation, each emission spectrum was normalized with varying concentrations of edelfosine (Fig 4B). Emission peaks at ~550 nm for NBD-Sph were observed at low concentrations of edelfosine; however, at concentrations exceeding 1 µM of edelfosine, the emission peaks shift to ~540 nm. The shift in the wavelength of the emission peaks indicates a change in the environment of the fluorophore (48, 49, 50), likely indicating the transition of NBD-Sph from being free in solution to being incorporated into mixed micelles with edelfosine. Initially, emission peaks at ~530 nm are observed, assigned to be NTD and NBD-Sph complexes (Fig 4C). As the edelfosine concentration exceeds 2–3 µM, the emission peaks shift toward 540 nm, suggesting that edelfosine effectively outcompetes NBD-Sph, causing NBD-Sph to form micelles. Similarly, for binding of substrates at higher concentrations than their CMC, this binding assay is rather complex because of edelfosine exceeding its

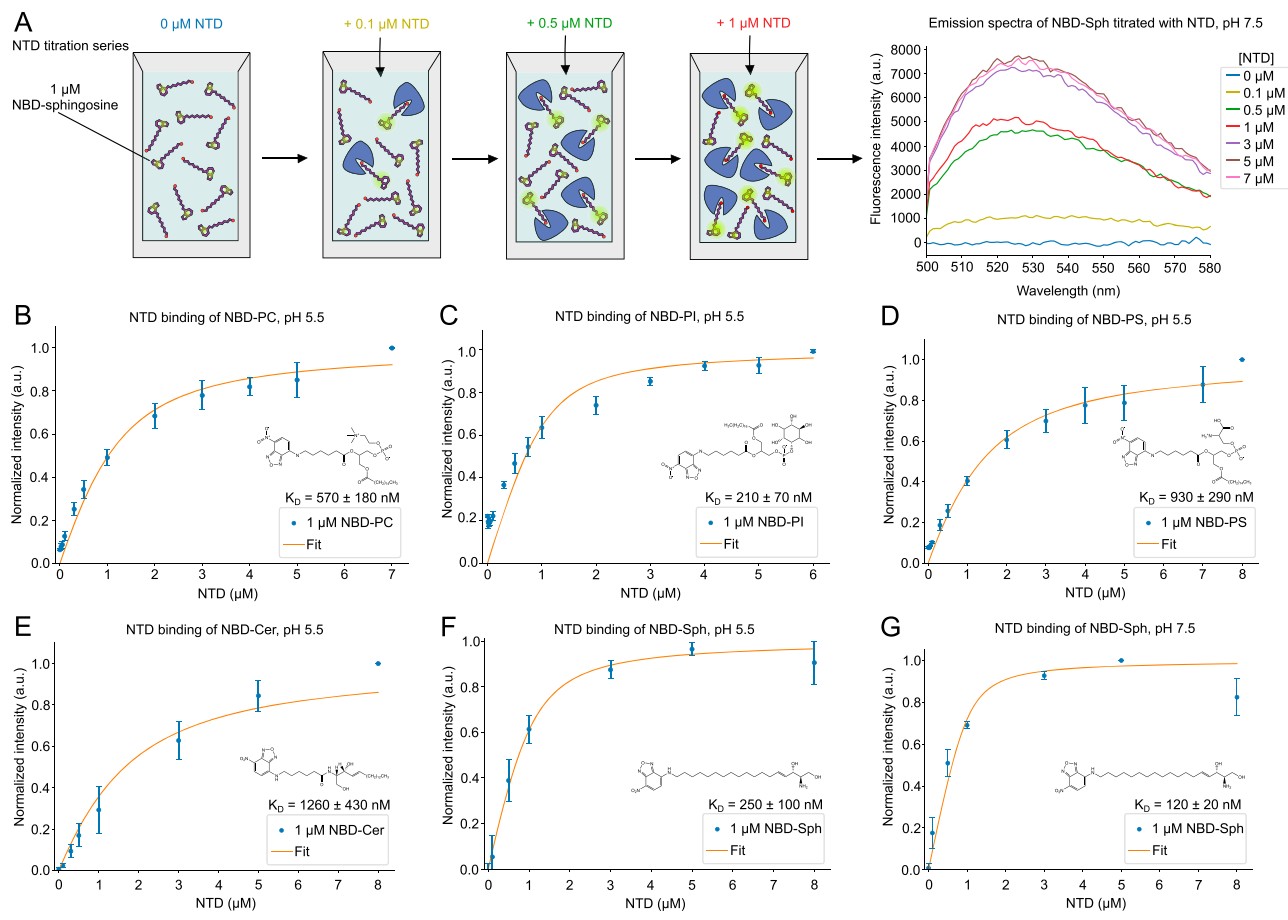

**Figure 3. NTD binding to phospholipids and sphingolipids.**
**(A)** Overview of the principle behind the fluorescence binding assay. The graph shows representative raw emission spectra in a range of 500 to 580 nm of NBD-sphingosine (NBD-Sph), measured with varying concentrations of the NTD. The excitation wavelength was 460 nm. Measurements were conducted using a 1 $\mu$M NBD-Sph solution in 50 mM Tris (pH 7.5), while titrating increasing concentrations of the NTD. After the addition of NTD, the solution was incubated for 10 min to ensure equilibrium was reached before the next measurement. **(B, C, D, E)** Fluorescence for NBD-phosphatidylcholine, NBD-phosphatidylinositol, NBD-phosphatidylserine, and NBD-ceramide, respectively, is shown as a result of binding to increasing concentrations of the NTD at pH 5.5. The measurements were conducted at an excitation wavelength of 460 nm and emission wavelength of 530 nm. For each lipid–protein complex, a dissociation constant ($K_D$) is determined and shown in the corresponding graphs. Data points show the mean ± SEM of three (n = 3) independent experiments. **(F, G)** Normalized fluorescence signal for NBD-Sph at an excitation wavelength of 460 nm and an emission wavelength of 530 nm shows increasing fluorescence intensity as a result of binding to increasing concentrations of NTD at pH 5.5 (F) or pH 7.5 (G). The $K_D$-values are determined and shown in the graphs. Data points show the mean ± SEM of three (n = 3) independent experiments.

CMC. Furthermore, Equation (1) does not account for NBD-Sph being able to participate in micelle formation. Because of these restrictions, the inhibitory constant for edelfosine is referred to as apparent, rather than a real, inhibitor constant ($K_I$). Equation (2) was used to determine the $K_I$ to ~ 50 nM based on a half-maximal inhibitory concentration (IC50) of 0.48 $\mu$M derived from Fig 4A. This suggests that the NTD binds edelfosine more strongly than NBD-Sph, explaining why edelfosine can out-compete NBD-Sph.

Based on the above observations, fixed amounts of edelfosine were added to NBD-Sph before titrating with the NTD in the second assay (Fig 4D). The measurements show that more NTD is needed to saturate the binding of NBD-Sph as the edelfosine concentration increases, suggesting that NBD-Sph and edelfosine compete for the binding pocket of the NTD. This observation is confirmed by the increase in apparent $K_D$-values of NBD-Sph from ~120 to ~210 nM as the edelfosine concentration was raised. Together, results from

both assays demonstrate that edelfosine is likely a ligand of NCR1 and that it binds tighter to the NTD than NBD-Sph.

### The NTD and NPC2 crosslink with photo-activatable and clickable lipid analogs

In a parallel approach, the substrate scope of the NTD and NPC2 was probed with photo-activatable and clickable (pac) lipids. Pac-lipids contain small chemical modifications for click chemistry reactions, with fluorophores such as AF647 (a structural analog of Alexa Fluor 647), and for UV-activated photo-crosslinking, respectively (45). In contrast to NBD-tagged lipids delivered in solution, pac-lipids such as pac-cholesterol (pacChol), pac-phosphatidylcholine (pacPC), pac-sphingosine (pacSph), and pac-ceramide (pacCer) were reconstituted in liposomal membranes (Fig 5A and B). As expected, the delipidated NTD crosslinks with pacChol (average +UV/−UV ratio = 2.11). Crosslinking of the NTD

**Table 1.** Overview of $K_D$-values for lipid–protein complexes.

| Lipid | pH | Protein | KD (nM) |
|---|---|---|---|
| NBD-C6-PC | 5.5 | NTD | 570 ± 180° |
|  |  | NPC2 | 492*° |
| NBD-C6-PI | 5.5 | NTD | 210 ± 70 |
|  |  | NPC2 | 10* |
| NBD-C6-PS | 5.5 | NTD | 930 ± 290° |
|  |  | NPC2 | 20 ± 10° |
| NBD-C6-Cer | 5.5 | NTD | 1,260 ± 430° |
|  |  | NPC2 | 90 ± 80° |
| NBD-Sph | 5.5 | NTD | 250 ± 100 |
|  |  | NPC2 | 580 ± 160 |
| NBD-Sph | 7.5 | NTD | 120 ± 20 |
|  |  | NPC2 | 530 ± 170 |

Results from reference 28 are indicated as *. Apparent $K_D$-values are marked with °.

with pacSph (+UV/−UV = 1.73) and, to a lesser extent, with pacCer and pacPC (+UV/−UV = 1.40 and 1.12), respectively, was also detected (Fig 5C and D).

To compare with the interaction profile of the NTD, crosslinking assays were also performed with purified, delipidated NPC2 (Fig S4A and B). In general, NPC2 displays higher +UV/−UV ratios, suggesting a more efficient lipid binding, likely because of its large hydrophobic cavity (27). NPC2 crosslinks well with all tested pac-lipids but, as for the NTD, most strongly with pacChol (+UV/−UV ratio = 1.91). Robust and consistent crosslinking with pacSph (+UV/−UV ratio = 1.64), pacCer (+UV/−UV = 1.67), and pacPC (+UV/−UV = 1.63) were observed. The crosslinking efficiency for cholesterol and sphingosine analogs was higher for the NTD, whereas crosslinking with ceramide and PC analogs seemed stronger with NPC2.

The consistent interaction of sphingosine with both the NTD and NPC2 observed in this study inspired us to examine whether the transfer of pacSph between the two proteins could be reconstituted in vitro. To this end, a protein-to-protein sphingosine transfer assay using solubilized, free pacSph was designed (Fig 6A). To correct for non-specific fluorophore background, a His-tagged green fluorescent protein (His-sfGFP) was used because it does not crosslink with pac-lipids (our own unpublished observation). As a control for specificity of transfer, the lipid binding domain of a mammalian STARD3 (His-STARD3-StART) was used because this protein was previously shown to interact with sphingosine in related work (51 Preprint), but it is not located inside lysosomes and, therefore, is not expected to interact with the NTD. His-tagged NPC2, His-sfGFP, or His-STARD3-StART were immobilized on NiNTA beads, preloaded with pacSph, and incubated with the NTD. After removal of the beads, the NTD was subjected to UV crosslinking and clicked with a fluorophore, and the resulting lipid–protein complexes were visualized by SDS–PAGE. Our results reveal increased crosslinking between pacSph and the NTD when pacSph was delivered by NPC2, but not when it was delivered by GFP or STARD3 (Fig 6B and C). Similar assays with pacCer did not yield positive results (Fig S5).

Together, these results support the specific transfer of pacSph from NPC2 to the NTD and indicate the ability of the yeast NPC system to export vacuolar sphingosine.

Lastly, the impact of "competitive" lipids on the crosslinking profile of NPC2 was investigated because the spectroscopy assays show that edelfosine competes with NBD-sphingosine binding to the NTD (Fig 4). Previous reports also show that edelfosine decreases the binding of radioactive cholesterol by the NTD (27). To explore this further, pac-lipid–containing liposomes were supplemented with fivefold excess of edelfosine; however, no noticeable inhibition of crosslinking was observed (Fig S6A and B). In a similar way, the effects of ergosterol on protein–lipid interaction were evaluated. Ergosterol does not visibly influence the crosslinking with pacChol and pacPC, whereas a trend toward increased crosslinking with pacSph and pacCer is observed (Fig S6C and D).

## Discussion

X-ray crystallography was used to solve structures of the NTD, from *S. cerevisiae*, bound to ergosterol and cholesterol (Table S1). The binding pocket of the two structures is virtually identical, with only K90 adopting a slightly different rotamer (Fig 2B) but still capable of interacting with the backbone of G194 to form a "ridge" across the binding pocket, creating the "sterol opening" and "water opening" (Fig 2D). The binding pocket itself is lined mainly with hydrophobic residues, in particular, F109 and F112 pi-stack with the ring system of a sterol for strong binding (Figs 1D and F and 2B). One of the polar residues, Q80, interacts with the hydroxyl group of the sterol at the deepest point of the binding pocket. Three regions that do change in position are loops on the face of the substrate binding pocket, of which "mobile loop 3" contains the "ridge" that spans the binding pocket (Fig 2C and D). The most displaced loop is "mobile loop 2" at the top of the "sterol opening." Together, "mobile loop 2" and "mobile loop 3" likely gate substrate entry and exit. The regions proposed to govern substrate accessibility of the NTD from hNPC1 differ from what is proposed here for the NTD (Fig S7A), particularly helix 3, helix 7, and helix 8 of the NTD from hNPC1, all at the top of "sterol opening," which excludes the ridge-forming mobile loop of the NTD (33).

Most of the residues conserved between the NTD and the NTD from hNPC1 are in the binding pocket (Fig S7A). Structurally, however, the substrate binding pocket of the NTD from yeast is less occluded, with 10.2 Å between K90 and I172, compared with 8.1 Å between the equivalent residues in the NTD of hNPC1 (Figs 2B and S7B). Even though the binding pocket of the NTD is less occluded, cholesterol seems to bind 1.0 Å deeper compared with the NTD from hNPC1 (Fig S7B). The observation of cholesteryl hemisuccinate in the tunnel of full-length NCR1 supports the capability of the NTD to also bind sterols that are modified at the hydroxyl group (14), whereas the NTD of hNPC1 cannot (33). Similarly, the binding pocket of yeast NPC2 is larger than that of human NPC2 (27), which coincides with the NTD being able to accept a variety of substrates from NPC2 (28). The spacious binding mode of the NTD might allow other substrates, like single-chain lipids, to fit in the binding pocket,

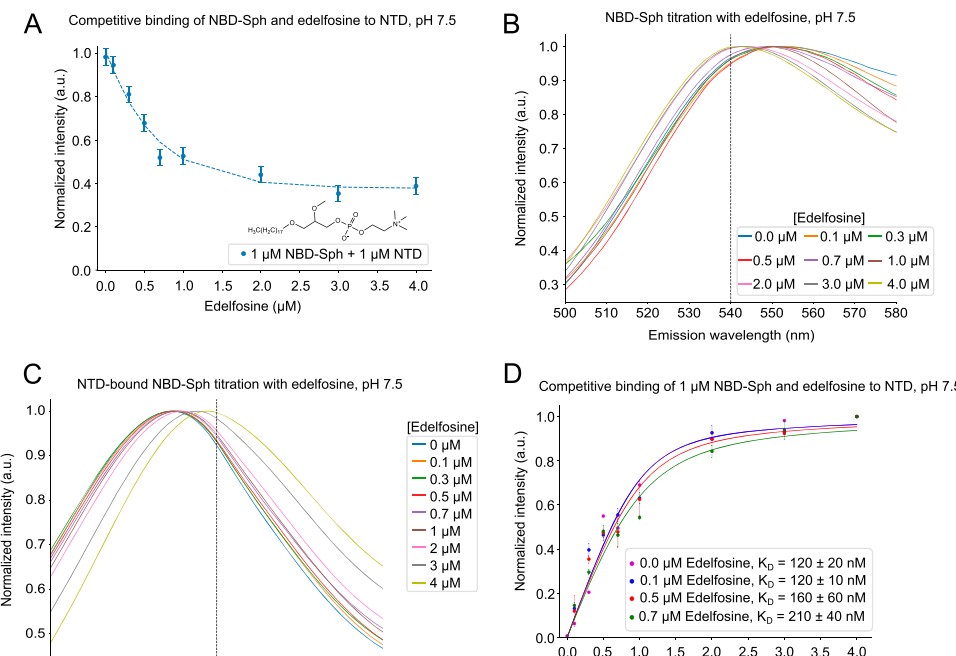

**Figure 4. Competitive binding of edelfosine and sphingosine to the NTD.**
**(A)** Normalized fluorescence of 1 μM NBD-sphingosine (NBD-Sph) mixed with 1 μM NTD, with concentrations ranging from 0 to 4 μM of edelfosine at pH 7.5. The excitation and emission wavelengths are 460 nm and 530 nm, respectively. The IC50 is estimated to be 0.5 μM. The data points show the mean ± SEM of three (n = 3) independent experiments. **(B)** Each emission spectrum of 1 μM NBD-Sph with varying edelfosine concentrations at pH 7.5 normalized to 1. The measurements were obtained using an excitation wavelength of 460 nm. The dashed line marks 540 nm. **(C)** Similar to (B) but with 1 μM NTD in the solution. **(D)** Normalized NBD-signal of 1 μM NBD-Sph incubated with either 0 (magenta), 0.1 (blue), 0.5 (red), or 0.7 (green) μM edelfosine at increasing concentrations of the NTD. The data points show the mean ± SEM of three (n = 3) independent experiments in the presence of edelfosine, whereas the data points without edelfosine show the mean of two (n = 2) independent experiments.

where the polar headgroup would be enveloped by the protein and the hydrophobic tail would be positioned toward the "sterol opening."

When comparing the NTD to the NTD from hNPC1, some unique features were observed. Only two of the five *N*-glycosylated residues of the NTD from hNPC1 (N122 and N185) (33) are conserved in the NTD (N123 and N178) (Fig S7A); however, all glycosylation sites are on the opposite face of the substrate binding pocket (Fig S1B and D), thus not interfering with sterol loading by NPC2. An ion coordination site, also opposite the substrate binding pocket, has not been described before (Fig 2A). It is not seen in the structures of full-length NCR1, and it does not seem to exist in the NTD from hNPC1. Although it is possible that this ion coordination was an artifact of the crystallization process, it is known that the vacuole stores transition metal and calcium ions (52, 53, 54), which the NTD seems capable of binding. Lastly, an extension of the substrate binding pocket downward, terminating next to the ion coordination site, is seen in the NTD (Fig S7C). The ~20 Å long pocket is solvent-accessible until a bottleneck (<1.4 Å) is formed by L108, F121, and F216.

When comparing the position of ergosterol in the crystal structure of the NTD at pH 6 of this study to that of the full-length NCR1 solved by cryo-EM (14), the sterol position differs. The ergosterol is bound ~4.5 Å deeper in the binding pocket of the crystal structure compared with the cryo-EM structures (Fig S7D). One explanation for this is that the crystal structure of the NTD represents a post-loading state, whereas the cryo-EM structures show the sterol approaching the transfer state (1).

The delipidated NTD did not crystallize, but crystals formed when C16 ceramide and edelfosine were added. Unfortunately, the data quality was not sufficient to confidently model these ligands, especially the long, flexible acyl chains that likely protrude from the binding pocket. However, these results prompted us to investigate the ability of the NTD to bind these compounds with biochemical assays. Previous in vitro assays have shown that the NTD from hNPC1 and hNPC2 can bind and transfer cholesterol and oxysterols (35, 37, 38). Binding assays show that yeast NPC2 binds not only cholesterol, ergosterol, dehydroergosterol (a fluorescent analog of ergosterol), and edelfosine (27), but also phospholipids and sphingolipids (28). This correlates with the larger binding pocket of NPC2 compared with human NPC2 (27). Here, it was assessed whether the NTD binds the same NBD-labeled substrates as NPC2.

In general, the intensities measured for NPC2 binding to the substrates are higher than the intensities for the NTD binding to the corresponding compounds (Figs 3A and S3A). This could be caused by the NBD group being in a more hydrophobic environment when bound by NPC2 compared with the NTD. Indeed, our previous MD simulation experiments (28) suggest that phospholipid acyl chains are buried inside the NPC2 binding pocket, which can flexibly adapt its size to the molecular volume of each ligand (sterol or phospholipid). Thus, an interesting future project could be to compare the orientation and flexibility of NBD-tagged lipids inside the binding pocket of NPC2 and the NTD using a combination of experimental and computational methods.

The binding of NBD-PS and NBD-Cer to NPC2 is more than an order of magnitude stronger compared with that of the NTD (Table 1). Based on this strong interaction, it is unlikely that NPC2 will hand over these substrates to the NTD. It can thus be postulated that NPC2 acts as a general lipid solubilizer in the vacuole and might have other interaction partners (14). The order of $K_D$-values of NBD-Sph strongly indicates a favorable transfer of the substrate from NPC2 to the NTD because both proteins bind the substrate, but

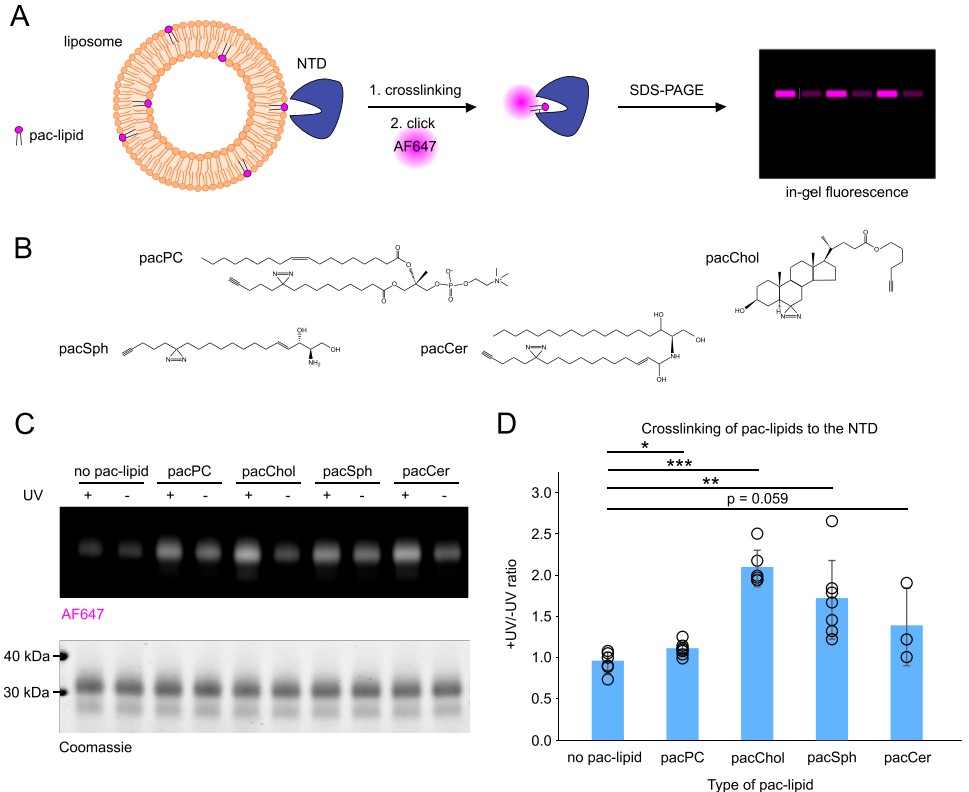

**Figure 5. NTD crosslinking with pac-lipids.**
**(A)** Schematic of the liposome crosslinking assay performed for experiments shown in (C) and quantified in (D). **(B)** Structures of pac-lipids used in the experiments. **(C)** Representative gel image shows the crosslinking profile of purified NTD. Liposomes containing 1.5 mol% of indicated pac-lipids (1 $\mu$M pac-lipid) were incubated with 1.5 $\mu$M NTD and UV-crosslinked. Protein–lipid complexes were subjected to the click reaction with AF647-picolyl azide and visualized by SDS–PAGE. In-gel fluorescence of AF647 was normalized to loaded protein in each well based on Coomassie staining. **(D)** Normalized in-gel fluorescence is quantified as the +UV/−UV signal ratio. N = at least three independent experiments.

the affinity for binding by the NTD is stronger (Table 1). A higher affinity of the NTD toward a given lipid ligand compared with that of NPC2 could indicate directed lipid transfer from NPC2 to the NTD.

The concentrations of NBD-PC and NBD-PS used in the fluorescence binding experiments exceeded the CMC values previously determined to be 0.116 and 0.165 $\mu$M, respectively (44). It can be speculated that the concentration of 1 $\mu$M NBD-Cer, used in the binding assays, also exceeded the CMC of this lipid analog. However, the CMC of NBD-Cer could not be measured because no discontinuity in the concentration-dependent fluorescence was found. In these cases, the determined $K_D$-values should be interpreted as apparent dissociation constants because the measured increase in fluorescence signal depends both on the binding equilibrium and on the thermodynamics of the micelle-to-monomer interconversion (28). In contrast, the concentrations of NBD-PI and NBD-Sph were lower than the CMCs determined to be 2.3 and 12.0 $\mu$M (Fig 3C and F), respectively (44); therefore, the measured fluorescence intensities for these lipids were most likely not affected by micelle formation. This is relevant because significantly different $K_D$-values can be determined if measured above and below the CMC of a lipid probe as previously shown for NBD-PC (28). The $K_D$ of NBD-PC binding to NPC2 is more than 100-fold smaller when the lipid concentration is below the CMC compared with above, but different photophysical mechanisms underlying the binding-induced fluorescence increase of NBD-lipids prevent a quantitative comparison of $K_D$-values below and above their CMC (28). Thus, the $K_D$-values for binding of NBD-PC, NBD-PS, and NBD-Cer to the proteins in this study cannot be directly compared with each other or with the $K_D$-values measured for NBD-PI and NBD-Sph, as for the latter lipids all binding studies were

carried out with concentrations below their respective CMC. Only in this case, the binding model used provides direct estimates of $K_D$-values because no pre-equilibrium between lipid monomers and formed micelles exists. For all binding measurements above the analog's CMC, the measured $K_D$-values should be considered as apparent dissociation constants.

Interestingly, we were able to measure binding of the non-fluorescent ligand, edelfosine, because of its competitive properties toward NBD-Sph indicated by the change in NBD fluorescence (Fig 4). As mentioned earlier, crystals of the NTD and edelfosine formed for X-ray crystallography, indicating the presence of the lipidated NTD. Together, these data support binding of edelfosine to the NTD. Previous research shows that ΔNCR1 yeast is resistant to the antifungal drug (41), indicating an interaction between edelfosine and NCR1 in the yeast vacuole. Others have reported that edelfosine targets the plasma membrane and ER by altering the sterol distribution in the membranes (55). Edelfosine might be internalized through endocytosis to the vacuole (41) followed by translocation to the ER. This strongly implies that edelfosine should exit the vacuole by binding to the NTD and passing through the tunnel of NCR1 to obtain its cytotoxic effect. In addition, these experiments confirm the binding of a phospholipid analog, adding to the broader spectrum of substrates that can bind to the NTD of NCR1.

It cannot be ruled out that the NBD-moiety affects the properties of the lipid analogs used here and, therefore, the binding properties that were measured. To consolidate our findings, a different technique, based on the photo-crosslinking with bifunctional pac-lipids in liposomes, was used. Among the selected pac-lipids,

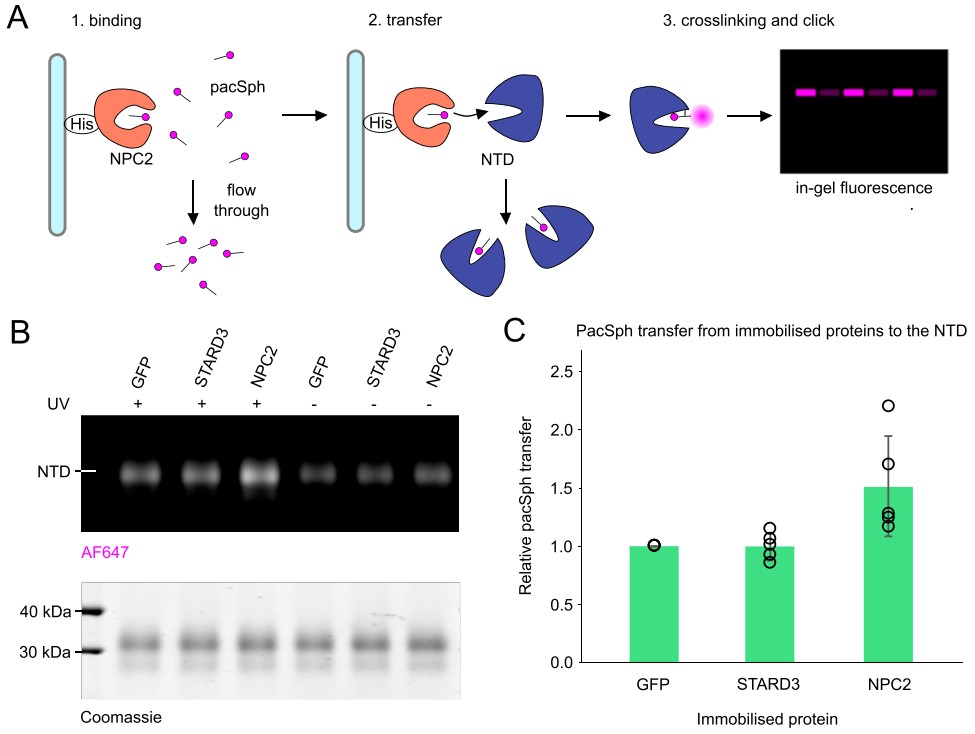

**Figure 6. Pac-lipid transfer assay to the NTD.**
**(A)** Schematic of the pacSph transfer assay performed for experiments shown in (B) and quantified in (C). **(B)** Representative gel image showing the fluorescently labeled pacSph-NTD complexes. Soluble pacSph was incubated with His-tagged GFP (normalization control), STARD3 (negative control), or NPC2. Unbound pacSph was washed away, and then, His-tagged proteins were incubated with soluble untagged NTD. After 1 h of incubation, the NTD was subjected to UV crosslinking and the click reaction with AF647. Crosslinked and stained protein–lipid complexes were resolved on SDS–PAGE. In-gel fluorescence of AF647 was first normalized to loaded protein. Then, +UV/−UV ratios were calculated. The +UV/−UV ratio of GFP was used as baseline (=value 1 for each experiment). **(C)** Quantification of normalized in-gel fluorescence. N = at least five independent experiments.

the NTD interacted mainly with cholesterol and sphingosine (Fig 5), in accordance with previous results in the mammalian NPC system (56), whereas NPC2 crosslinked with similar intensity with all tested ligands (Fig S4), suggesting a broader lipid binding function. These results correspond to the NBD-lipid assays, even though the crosslinking method is generally a more qualitative approach compared with fluorescence spectroscopy. The broad substrate scope of NPC2 could explain the more pronounced phenotypes in yeast when NPC2 is deleted, such as sterol accumulation, reduced vacuolar fusion, loss of raft-like domains in the limiting membrane, and, ultimately, vacuolar fragmentation (27, 57, 58). Also, the smaller number and more severe phenotype of patients with defective hNPC2 compared with hNPC1 could point toward a higher lethality because of the involvement of hNPC2 in multiple lipid trafficking pathways (59).

When examining the effects of membrane composition on lipid binding, it seems that supplementing liposomes with excess ergosterol increased crosslinking of NPC2 with pacSph and, to a lesser extent, pac-Cer (Fig S6C and D). This suggests a cooperative effect between ergosterol and sphingolipids in this in vitro setup. It can be speculated that the increased crosslinking with sphingolipid analogs could be caused by increased membrane recruitment and longer dwell time at sterol- and sphingolipid-enriched membrane regions (60). It was previously shown that sterol transfer by hNPC2 is slightly increased by ceramide and greatly promoted by lysobisphosphatidic acid (61, 62). However, no noticeable effect of edelfosine on the crosslinking with individual pac-lipids was observed (Fig S2A and B), possibly because of the low sensitivity of the crosslinking assay toward inhibitory effects.

The NTD from hNPC1 has been implicated in exporting lysosomal sphingosine (17, 56). Here, the protein-to-protein transfer assay

with photo-crosslinkable lipids revealed that pacSph was transferred from NPC2 to the NTD from yeast but not to STARD3 (Fig 6), a sphingosine interactor from mammals (51 *Preprint*). A recent study observed that NCR1 physically interacted with the ceramide synthase complex (63), which would be especially favorable for the rapid channeling of lysosomal sphingosine into ceramide synthesis, thus fuelling transport along its concentration gradient. Attempts to perform this assay with pacCer in solution did not yield a positive result, possibly because of the poor solubility of pacCer in buffer, which could prevent efficient loading of the His-tagged proteins (Fig S5).

In conclusion, the combination of X-ray crystallography and biophysics methods enables us to start uncovering the substrate scope of the NPC system from *S. cerevisiae*. We find that both NPC2 and the NTD interact with a wide range of structurally diverse lipids. Besides sterols, the yeast NPC system seems capable of trafficking various lipids to the limiting membrane of vacuoles. Future efforts should be directed toward elucidating the relative kinetics of multiligand transfer by the NPC proteins in yeast, the regulation of such transport, and the consequences of its defects on cellular growth and function.

# Materials and Methods

### Yeast cultivation and expression of the NTD

The DNA sequence of the N-terminal domain of the NCR1 protein (residues 1–249, UniProt: Q12200) was introduced into an expression construct based on p423-GAL1, with a C-terminal thrombin cleavage

site and a deca-histidine tag for purification (64). Transformed *S. cerevisiae* (strain DSY-5; Gentaur) was grown in baffled shaker flasks for 30 h and harvested after a 22-h induction using galactose (65). Harvested cells were washed in cold water, spun down, and stored at –80°C.

## Purification of the NTD for crystallization and assays

Cells were thawed in lysis buffer (600 mM NaCl, 100 mM Tris–HCl, pH 7.5) supplemented with 1.2 mM PMSF, while gently stirring for 30 min. The cells were lysed by agitation with 0.5-mm glass beads (Biospec Products). The cell lysate was removed from the glass beads by filtering and then centrifuged at 9,000 rpm (Sorvall Lynx 6000 centrifuge, F9-6x1000 LEX rotor; Thermo Fisher Scientific) at 4°C for 20 min to pellet cell debris. The cell-free extract was ultra-centrifuged at 42,000 rpm (Sorvall wX+ UltraSeries centrifuge, T-647.5 rotor; Thermo Fisher Scientific) at 4°C for 1 h to pellet membranes and cell debris. The supernatant was filtered and loaded onto a 5 ml IMAC NiNTA HP column (Cytiva) pre-equilibrated with wash buffer (500 mM NaCl, 10% [vol/vol] glycerol, 20 mM imidazole, 50 mM Tris–HCl, pH 7.5). Next, contaminant proteins were washed away with 50 ml of W70 buffer (wash buffer with 70 mM imidazole).

If endogenous lipid was retained, 45 ml of G20 buffer (200 mM NaCl, 20 mM imidazole, 20 mM Tris–HCl, pH 7.5) was passed through the column and 175 units of thrombin (Avantor) were added to 5 ml of G20 to circulate over the column at 4°C, overnight. Because the His-tag had been cleaved off, the NTD was eluted with 15 ml G40 buffer (200 mM NaCl, 40 mM imidazole, 20 mM Tris–HCl, pH 7.5) and was concentrated to <500 $\mu$l in a 20-ml 10-kD MWCO concentrator (Sartorius Vivaspin). Aggregates were removed by filtering the concentrated sample with a 0.22-$\mu$m PVDF membrane spin column (Durapore; Merck). The flow-through was injected onto an S75 Increase 10/300 GL column (Cytiva), pre-equilibrated with G-buffer (150 mM NaCl and 20 mM MOPS, pH 6.5). The peak fractions were pooled and concentrated with a 500-$\mu$l 10-kD MWCO concentrator to 3.3 mg/ml for assays with NBD-lipids and 1 mg/ml for assays with pac-lipids.

If the endogenous lipid had to be removed, the same procedure as above was applied but with two additional wash steps. Before the step of washing with G20 buffer, 100 ml of W70S buffer (W70 buffer supplemented with 100 mM methyl-$\beta$-cyclodextrin) was passed through the column, followed by 50 ml of W70 buffer, purified as described above, and used in substrate binding assays. For crystallography, the endogenous lipid was removed with methyl-$\beta$-cyclodextrin and exchanged with 50 ml of W70 buffer containing 50 $\mu$M lipid of choice, (cholesterol, ergosterol, C16 ceramide, or edelfosine), dissolved in 2 CMC of *n*-dodecyl-$\beta$-D-maltoside (DDM) detergent. The detergent was removed by washing with 100 ml of W70 buffer, and the rest of the purification was followed as described above.

## Yeast cultivation and expression of NPC2

The gene encoding the NPC2 protein (residues 1–173, UniProt: Q12408) of *S. cerevisiae* was cloned into the same type of the expression vector as described for the NTD. Cell growth and induction of homologous overexpressed protein were the same as for the

NTD. Harvested cells were washed in cold water, spun down, and stored at –80°C.

## Purification of NPC2 for assays

The purification protocol for the NPC2 was identical to that of the NTD, except the G-buffer used for size-exclusion chromatography (200 mM NaCl, 20 mM Tris–HCl, pH 7.5). To delipidate NPC2 with methyl-$\beta$-cyclodextrin, the same procedure was applied as described above for the NTD. The peak fractions were pooled and concentrated with a 500-$\mu$l 10-kD MWCO concentrator, either to 2.2 or to 1 mg/ml for biochemical assays with NBD-lipids and pac-lipids, respectively.

## Deglycosylation of the NTD

0.5–5 mg/ml of enzyme and 1 mg/ml of the purified NTD were incubated on a rotating table at 4°C with one of five deglyco-sylases: PNGase F (hydrolyzes the bond between asparagine and the first *N*-acetylglucosamine), Endo H (hydrolyzes the glycosidic linkage between the two core *N*-acetylglucosamines), Endo F1 (cleaves hybrid but not complex oligosaccharides), Endo F2 (leaves one *N*-acetylglucosamine attached to asparagine), and Endo F3 (does not cleave high mannose or hybrid glycan chains) (66, 67). For controls, the NTD without enzymes and the enzyme alone were also prepared. SDS loading dye was added to the samples after 1 h, boiled for 1 min, and visualized by SDS–PAGE.

## NTD crystallography, data collection, processing, and building

The purified NTD bound to either ergosterol or cholesterol was evaluated with crystallization screens. Conditions containing 20–300 mM ZnSO$_4$, 45–48% (vol/vol) PEG200, and MOPS/MES at pH 5.5–6.5 yielded crystals at 19°C and micro-seeding of the NTD at 8 mg/ml. Optimization of initial crystal hits was done in 50 mM ZnSO4, 42% PEG200, and 100 mM MES, pH 6, with the addition of the Hampton additive screen in 96-well plates. A Mosquito Robot (Mosquito Xtal3; SPT Labtech) was used to add 200 nl of the NTD at 16 mg/ml–200 nl of mother liquor, and the trays were incubated at 4°C. After 8 d, crystals appeared and were left to keep growing. The NTD bound to ergosterol produced crystals with 4% (vol/vol) acetonitrile as an additive, and the NTD bound to cholesterol produced crystals with 3% vol/vol 2-methyl-2,4-pentanediol (MPD) as an additive. Additional cryoprotectant was not added before freezing in liquid nitrogen.

Final datasets were collected at the MAXIV beamline (BioMAX) using an EIGER DECTRIS 16 M detector. XDSapp v3.1.9 was used to process the data, and the NTD from NCR1 (PDB ID: 6R4L) was used for molecular replacement with the Phaser module in PHENIX v1.20 (68, 69, 70). Extensive glycosylation resulted in high solvent content and translational non-crystallographic symmetry. MR solutions were obtained in space group 18 (P2$_1$22$_1$). Iterative rounds of model building and refinement were done in Coot v0.9.8.7 (71) and phenix.refine, respectively. MolProbity statistics were used to guide model building (Table S1) (72). The final models and maps were

validated and deposited on the PDB OneDep server (73) and raw data on XRDa (74).

## Fluorescence binding assays

Fluorescence binding assays were conducted using NBD-PC, NBD-PS, NBD-Cer, and NBD-Sph (Avanti Polar Lipids), and NBD-PI (Prof. Bütikofer), dissolved in ethanol. A solution of 1 $\mu$M NBD-lipid in either MES buffer (200 mM NaCl, 50 mM MES, pH 5.5) or Tris buffer (50 mM Tris, pH 7.5) was transferred to 0.1-cm-thick Quartz cuvettes. NBD-lipids were excited at 460 nm, and the emission spectra were recorded within a range of 490–600 nm. Either the NTD or NPC2 was titrated in increasing concentrations to the sample cuvette. The sample solution, not exceeding 1% ethanol, was mixed with protein and incubated for 10 min. Measurements were conducted with an ISS Chronos spectrofluorometer (Urbana-Champaign, IL) with 0.5 mm slit width and no polarization. Emission spectra for NBD-lipids were measured as a control and subtracted from the sample emission spectra. The control cuvette was measured every time the sample was measured to correct the data for bleaching of NBD-Sph.

For measuring the ability of edelfosine to outcompete NBD-Sph, 1 $\mu$M NBD-Sph and 1 $\mu$M NTD were mixed to form the interaction complex before adding increasing amounts of edelfosine. As a control, edelfosine was titrated into a cuvette containing exclusively NBD-Sph in buffer. To measure the competitive binding between NBD-Sph and edelfosine, 1 $\mu$M NBD-lipid in Tris buffer was mixed with either 0.1, 0.5, or 0.7 $\mu$M edelfosine in the sample cuvette before titration with the NTD.

## CMC measurements

The CMC of NBD-Sph was measured by adding increasing concentrations of NBD-Sph to a cuvette containing MES buffer. Between measurements, the solution was mixed and incubated for 10 min. The highest measured concentration of NBD-Sph was 30 $\mu$M.

As edelfosine is not fluorescent, a different approach was used to measure its CMC. The hydrophobic dye, Sudan Black B, was used to stain micelles. The absorption of solutions consisting of MES buffer, Sudan Black, and increasing concentrations of edelfosine was measured using a spectrophotometer (Hitachi U-2010).

## Analysis of spectroscopy data

Before determination of dissociation constants, the measurements were normalized. The protein–lipid binding measurements were normalized to the highest intensity in each dataset. For the competition assay, the data were normalized to the lowest intensity in the dataset to be able to fit all data to Equation (1), the full model for one-site binding for the saturated fraction, $f$ (27):

$$\frac{[RL]}{R_t} = \frac{R_t + K_D + L_t - \sqrt{(R_t + K_D + L_t)^2 - 4 \cdot R_t \cdot L_t}}{2 \cdot R_t} = f \tag{1}$$

Here, [RL] is the concentration of the receptor–ligand complex, $R_t$ is the total concentration of the receptor, and $L_t$ is the concentration of the ligand. Because the protein is titrated, the lipids are acting as the receptor, whereas the protein counts as the ligand.

To determine the dissociation constant of edelfosine (referred to as $K_I$) in the competition assay, it can be assumed that the binding is simple and that all added ligand is bound. In this case, the IC50 can be used to calculate a $K_I$ using Equation (2) on normalized data:

$$K_I = IC50 \cdot \frac{K_D}{K_D + [L]} \tag{2}$$

The $K_D$ is the dissociation constant of the ligand, [L] is the concentration, and IC50 is the concentration of the inhibitor, where half of the ligand is inhibited.

To determine the CMC, the sigmoid curve was fitted and adjusted to the normalized data:

$$f(x) = \frac{l}{1 + e^{-k \cdot (x - x_0)}} + b \tag{3}$$

Here, l is the scaling of the y-axis, $x_0$ represents the CMC, k determines the sharpness of the fitted curve, and b is the starting position on the y-axis.

## Liposome preparation

Lipids from stock vials were dissolved in chloroform and mixed at the desired molar ratio. For in vitro crosslinking, most liposomes consisted of 88.5 mol% DOPC as carrier lipid, 10 mol% DOPS, and 1.5 mol% respective pac-lipid. The pac-lipids were incorporated into liposomes at low concentrations (1.5 mol%) to ensure homogenous distribution and avoid micelle formation. The solvent was evaporated in a rotary evaporator (Hei-VAP Core, Heidolph) for 30 min at 42°C at 150 rpm to create a thin lipid film. Lipid films were hydrated with filtered and degassed PBS (137 mM NaCl, 2.7 mM KCl, 10 mM $Na_2HPO_4$, 1.8 mM $KH_2PO_4$, pH 7.4). Solutions of hydrated lipids (with a total lipid concentration of 2 mM) were subjected to five freeze–thaw cycles in liquid nitrogen and warm water (42°C) to facilitate breakup of large structures and support formation of smaller multilamellar vesicles. Lipid solutions were stored at –20°C. Before use, liposomes were extruded by passing 20x through a 100-nm pore size polycarbonate filter (Nuclepore; Whatman) using a hand extruder (Avanti Polar Lipids). Extruded liposomes were stored at 4°C, protected from light, and used within 1–2 d.

## In vitro crosslinking assay

Extruded liposomes were diluted in PBS until a final lipid concentration of 60 $\mu$M, including 1 $\mu$M pac-lipid. Purified proteins were added until a final concentration of 1.5 $\mu$M in a reaction volume of 100 $\mu$l. Protein–liposome mixtures were incubated in 0.5-ml Eppendorf tubes for 30 min at 37°C at 400 rpm (Comfort thermo-mixer, Eppendorf), then crosslinked (or not) by exposure to 365-nm UV light for 15 min at 4°C using a 100-W mercury lamp (Blak-Ray B-100AP). After crosslinking, the click reaction was performed by adding 0.7 $\mu$l of freshly prepared click mix into the sample tubes.

The final concentrations of click reagents in sample tubes were 80 $\mu M$ $CuSO_4$ (Merck), 3 $\mu M$ TBTA (Merck), 3 $\mu M$ AF647-picolyl azide (Jena Bioscience), and 80 $\mu M$ ascorbic acid (Merck). Fresh stock of ascorbic acid solution in water was prepared every time and added to the click mix immediately before starting the click reaction. Click reactions were allowed to proceed for 2 h at 37°C at 400 rpm with the thermomixer. Then, samples were concentrated in a vacuum evaporator for 30 min set to 30°C and supplemented with 4x Lämmli buffer (250 mM Tris, pH 6.8, 9.2% SDS, 40% glycerol, 0.2% bromophenol blue, 100 mM DTT). Samples were boiled for 5 min at 95°C, and clicked protein–lipid complexes were resolved by SDS–PAGE. After extensive destaining in a destain solution (65% water, 25% isopropanol, and 10% acetic acid), the AF647 (a structural analog of Alexa Fluor 647) and Coomassie images were acquired using the LI-COR Odyssey imaging system. Data were evaluated by subtracting background fluorescence, followed by comparing the AF647 signal between crosslinked and non-crosslinked samples, normalized to protein loading based on Coomassie staining. The resulting values are described as +UV/−UV ratios. Statistics were calculated with the Mann–Whitney $U$ test.

### In vitro pacSph transfer assay

Purified His-tagged proteins (5 $\mu M$ of sfGFP, STARD3, or NPC2) were incubated with 10 $\mu M$ pacSph in PBS for 60 min at 25°C at 500 rpm (Comfort thermomixer, Eppendorf). Next, the His-tagged proteins were captured on NiNTA beads and washed 5x with PBS to remove unbound pacSph. NiNTA beads with bound proteins were then incubated with 2.5 $\mu M$ of the untagged NTD for 60 min at 25°C at 500 rpm with the thermomixer. After removal of the beads, the NTD was subjected (or not) to UV crosslinking. After crosslinking, the click reaction was performed as described above and samples were resolved on SDS–PAGE. For data analysis, the background was subtracted and the +UV/−UV ratios were quantified for each sample as described previously. After that, the +UV/−UV ratio of sfGFP, incapable of binding sphingosine, was set as baseline (=1) to compare the pacSph transfer capability of STARD3 and NPC2 with the NTD. Statistics were calculated with the Mann–Whitney $U$ test.

### Figure preparation

All figures were annotated in Inkscape, and graphical representations of the structures were made in ChimeraX v1.71 (75). Graphs were created in GraphPad Prism 10, whereas CaverWeb v1.2 (76) was used to obtain dimensions of the extended pocket.

## Data Availability

Atomic models and experimental maps have been deposited in the Protein Data Bank (PDB). The NTD with ergosterol accession number is PDB 9F40. Raw data are available at XRDa as XRD-00255. The NTD with cholesterol accession number is PDB 9F41. Raw data are available at XRDa as XRD-00256.

## Supplementary Information

## Acknowledgements

We thank Prof. Bütikofer (University of Bern) for C6-NBD-PI. We recognize beamlines I24 and I04 at the Diamond Light Source and beamline BioMAX at the MAX IV Laboratory, where X-ray data were collected, and DESY-PETRA III for crystal screening. We acknowledge access to the computational infrastructure at the Center for Structural Biology at Aarhus University. We thank Julian Funk at Heidelberg University, for technical assistance. This work was supported by the Danish Council for Independent Research (Grant Agreement No. 0135-00032B to BP Pedersen and 2032-00139B to D Wüstner), the Carlsberg Foundation (Grant Agreement No. CF19-0127), the European Research Council (Grant Agreement No. 637372) to BP Pedersen, and the Deutsche Forschungsgemeinschaft (DFG, German Research Foundation, project number JA 3315/1-1 to D Jamecna and project number 278001972—TRR 186 to D Höglinger).

### Author Contributions

L Nel: conceptualization, formal analysis, investigation, methodology, and writing—original draft, review, and editing.
K Thaysen: formal analysis, investigation, methodology, and writing—original draft, review, and editing.
D Jamecna: formal analysis, investigation, methodology, and writing—original draft, review, and editing.
E Olesen: formal analysis, investigation, and methodology.
M Szomek: investigation and methodology.
J Langer: investigation and methodology.
KM Frain: investigation and methodology.
D Höglinger: conceptualization, formal analysis, supervision, funding acquisition, methodology, project administration, and writing—original draft, review, and editing.
D Wüstner: conceptualization, formal analysis, supervision, funding acquisition, methodology, project administration, and writing—original draft, review, and editing.
BP Pedersen: conceptualization, formal analysis, supervision, funding acquisition, methodology, project administration, and writing—original draft, review, and editing.

### Conflict of Interest Statement

The authors declare that they have no conflict of interest.

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
