## [Reviewer comments · Life Science Alliance]

Life Science Alliance

Structural and biochemical analysis of ligand binding in yeast Niemann-Pick type C1-related protein

Lynette Nel, Katja Thaysen, Denisa Jamecna, Esben Olesen, Maria Szomek, Julia Langer, Kelly Frain, Doris Hoeglinger, Daniel Wustner, and Bjørn Pedersen

DOI: 10.26508/lsa.202402990

Corresponding author(s): Bjørn Pedersen, Aarhus University and Daniel Wustner, University of Southern Denmark

Review Timeline:

Submission Date:	2024-08-11
Editorial Decision:	2024-09-13
Revision Received:	2024-10-10
Editorial Decision:	2024-10-11
Revision Received:	2024-10-15
Accepted:	2024-10-16

Transaction Report:

September 13, 2024

Re: Life Science Alliance manuscript #LSA-2024-02990

Dr. Bjørn Panyella Pedersen
Aarhus University
Department of Molecular Biology and Genetics
Gustav Wieds Vej 10
MBG-AU
Aarhus C, Danmark 8000
Denmark

Dear Dr. Pedersen,

Thank you for submitting your manuscript entitled "Structural and biochemical analysis of ligand binding in yeast Niemann-Pick type C1-related protein" to Life Science Alliance. The manuscript was assessed by expert reviewers, whose comments are appended to this letter. We invite you to submit a revised manuscript addressing the Reviewer comments.

Thank you for this interesting contribution to Life Science Alliance. We are looking forward to receiving your revised manuscript.

Sincerely,

B. MANUSCRIPT ORGANIZATION AND FORMATTING:

Reviewer #1 (Comments to the Authors (Required)):

This is a very nice piece work that details the yeast NCR1 (analogue of NPC1 from mammals) and NPC2 lipid transport system. The N-terminal domain (NTD) of NCR1 is shown to bind ergosterol and cholesterol (structurally similar by the crystallography done herein). A range of lipid-binding assays is included to confirm binding of lipid analogs os cholesterol, sphingosine and ceramide. A detailed mechanistic model is provided. This is a major advance in the field.

Overall, the data strongly support the conclusions throughout the paper and there are not any major concerns from this reviewer. I do have some minor concerns for improvement:

- I leave decision to editor and authors but I'd much prefer a much more detailed figure legend for figures 1 and 2. This is an opportunity to guide the reader through the structure and the details highlighted in each panel.
- For figure 3 and table 1, the authors may want to check the significant figures to report for Kd. Based on the protein and lipid concentrations used it may be better to report 2 significant figures instead of 3? For instance, 930 +/- 300 as opposed to 928 +/- 280 nM. The authors can check based on the input values and significant figures of lipid and protein concentrations known. Same for the significant figures reported in figure 4.
- For figure 5C, N = at least 4. It looks like the pacCer may only be 3 points but cannot completely check if overlapping. Please check if N = 3 or 4 for this lipid.
- In figure 5A cartoon, is it known/clear that the lipid is pulled out by the NTD or is the NTD still associated with the liposome?
- For figure S7D it lists N = at least 4 but for the pacChol and pacChol + arg it may only be 3 data points shown (please check).
- As with reviewer 2's comments, the edelfosine experiments could be removed as it is not a strong fit with the rest of the manuscript. The sturcture/pictures of pac-lipids should also be shown in the manuscript.

Reviewer #2 (Comments to the Authors (Required)):

This manuscript addresses the comparison between the N-terminal domains of mammalian cholesterol transport protein, NPC1, and its yeast homolog, NCR1. The NPC2 proteins of both species are also investigated. The manuscript uses x-ray crystallography and several biophysical methods with sterol and lipid analogs to characterize the similarities and differences of these lipid transporters. A major conclusion is that the yeast NPC2/NCR1 transport system has broader substrate specificity than the human proteins.

I found the overall paper to be interesting and significant, but I found several issues that I think need to be addressed in a revised manuscript.

1. There are some serious issues with the assays for binding of lipid derivatives to the N-terminal domain of NCR1 and NPC2.
 - A. I don't see any value in measurements of binding constants when using concentrations above the critical micelle concentration.
 - B. I don't see the value of the edelfosine experiments. It is not shown to bind in the binding pocket, and it could be a non-competitive inhibitor. This seems to be unrelated to the rest of the manuscript, and I would recommend deletion of this section.
2. The structures of the pac-lipids need to be shown.
3. The increase in fluorescence upon incubation with the pac-lipids seems extremely small. What does it mean to have a 2-fold increase in fluorescence? What percent of the NTD proteins have been labeled??
4. They show models of different orientations of the NBD group binding to the NTD of NCR1 vs. NPC2. They mention in the Discussion that it might be interesting to check this. I believe it is essential for interpreting these data.

Figure S1A needs a more complete figure legend. It is very hard to understand as written.

Referee Cross-Comments

I agree with Reviewer 1 that this is a valuable contribution.

I would like the authors to take my review into consideration before returning a final version of the manuscript.

Reply to Reviewers.

(our answer in blue)

Reviewer #1 (Comments to the Authors (Required)):

This is a very nice piece of work that details the yeast NCR1 (analogue of NPC1 from mammals) and NPC2 lipid transport system. The N-terminal domain (NTD) of NCR1 is shown to bind ergosterol and cholesterol (structurally similar by the crystallography done herein). A range of lipid-binding assays is included to confirm binding of lipid analogs of cholesterol, sphingosine and ceramide. A detailed mechanistic model is provided. This is a major advance in the field.

We thank the reviewer for the positive feedback.

Overall, the data strongly support the conclusions throughout the paper and there are not any major concerns from this reviewer.

I do have some minor concerns for improvement:

- I leave decision to editor and authors but I'd much prefer a much more detailed figure legend for figures 1 and 2. This is an opportunity to guide the reader through the structure and the details highlighted in each panel.

We have expanded upon the existing figure legends. The changes have been made in the text and indicated in purple below:

Figure 1: Structures of the NTD bound to sterols

(A) Structure of NCR1 (PDB ID: 6R4L) in grey with the NTD in color. NTD loading from NPC2 (PDB ID: 6R4N) is followed by transfer, transport and integration by NCR1 into the vacuole membrane. The NTD has seven α -helices, which are interrupted by two β -sheets between α 4 and α 5. After α 7, the third β -sheet connects the NTD with the long loop leading to the M1 transmembrane helix. The color gradient starts as blue at the N-terminus and transitions to red at the C-terminus. **(B)** Secondary structure elements of the NTD. **(C)** Chemical structure and density of ergosterol in the binding pocket of the NTD. The double bond between C22 and C23 makes the aliphatic tail of ergosterol rigid and can be seen within the continuous density in orange surrounding the molecule. **(D)** Residues surrounding ergosterol in the binding pocket of the NTD are mostly hydrophobic, except for Q80, N87, T113 and S196. The hydroxyl-group of ergosterol is coordinated by Q80, 3.0 Å away. **(E)** Chemical structure and density of cholesterol in the binding pocket of the NTD. In cholesterol, the double bond is lacking between C22 and C23 and makes the aliphatic tail more flexible, as can be seen in the surrounding discontinuous orange density. **(F)** Residues surrounding cholesterol in the binding pocket of the NTD are the same as for ergosterol. The hydroxyl group of cholesterol is closer to Q80, 2.8 Å away.

Figure 2. Structural features of the NTD

(A) Ion-coordinating residues are found at the back opposite face of the substrate binding pocket. Density surrounding residues E67, N107, H111, D218 and the zinc ion are shown in orange. **(B)** Superposition of NTD structures bound to ergosterol and cholesterol. The sterols and residues of both binding pockets are overlaid and show the same positioning between the two structures. **(C)** When overlaying the NTD bound to ergosterol and cholesterol, three "mobile loops" are displaced when comparing the two structures. These mobile loops cover the substrate binding pocket at the bottom, middle and top. ~~with different positions when comparing the NTD bound to ergosterol and~~

~~cholesterol.~~ (D) Residues that form the “ridge” include N87, K90 and G194 - with G194 being on “mobile loop 3” - and ~~that~~ likely govern substrate accessibility to the binding pocket. The ridge ~~also~~ divides the pocket into a “sterol-opening”, with the aliphatic tail of the sterol being visible, and a “water-opening”, which houses the hydroxyl group of the sterol.

- For figure 3 and table 1, the authors may want to check the significant figures to report for K_d. Based on the protein and lipid concentrations used it may be better to report 2 significant figures instead of 3? For instance, 930 +/- 300 as opposed to 928 +/- 280 nM. The authors can check based on the input values and significant figures of lipid and protein concentrations known. Same for the significant figures reported in figure 4.

This is a good point. Figures 3, S3 and 4 as well as Table 1 have been updated. The main text has also been updated accordingly.

- For figure 5C, N = at least 4. It looks like the pacCer may only be 3 points but cannot completely check if overlapping. Please check if N = 3 or 4 for this lipid.

Thank you very much for pointing out the errors in Ns. Indeed, there are only 3 data points for the pacCer and pacChol + erg experiments shown on the respective graphs – now both legends of Figure 5 and Figure S7 state the numbers correctly (see related comment below).

-In figure 5A cartoon, is it known/clear that the lipid is pulled out by the NTD or is the NTD still associated with the liposome?

In theory both possibilities can occur, depending on at which moment the UV crosslinking happens – either the lipid is already taken up by the protein and pulled out of the liposome, or it is during the initial stages when the NTD interacts with the membrane but has not yet extracted the lipid from the liposome. In this case, the lipid might only crosslink with some surface residues (on the loop, for example) of the protein and then either be pulled out (when it is relatively hydrophilic lipid such as sphingosine) or it can stay in the membrane and anchor the NTD to the liposome, when it is a more hydrophobic ligand like ceramide or cholesterol. We do not know what portion of the NTD crosslinks at which residue(s), and it is very likely that both cases occur in every test tube. However, upon the addition of Lämmli buffer and boiling of the samples prior to loading them on gel, liposomes disintegrate and only protein-lipid complexes run on the SDS-PAGE.

In the discussion, the sentence

“To consolidate our findings, a different technique, based on the extraction of bi-functional pac-lipids from liposomes, was used.”

was changed for a more accurate description:

“To consolidate our findings, a different technique, based on the photocrosslinking with bi-functional pac-lipids in liposomes, was used.”

-For figure S7D it lists N = at least 4 but for the pacChol and pacChol + arg it may only be 3 data points shown (please check).

Thank you very much for pointing out the errors in Ns. Indeed, there are only 3 data points for the pacCer and pacChol + erg experiments shown on the respective graphs – now both legends of Figure 5 and Figure S7 state the numbers correctly.

-As with reviewer 2's comments, the edelfosine experiments could be removed as it is not a strong fit with the rest of the manuscript. The structure/pictures of pac-lipids should also be shown in the manuscript.

We find the edelfosine experiments relevant and valuable, but agree we do not sufficiently explain this in the manuscript. To convince the reader and reviewer of the relevance of these experiments we have added a short paragraph in the Discussion in lines 338-349. The new text emphasizes the role of NCR1 and NPC2 in exporting edelfosine from the vacuole, which could underline the observed edelfosine resistance in yeast lacking the NPC transport system. We feel that our results on edelfosine binding by these proteins provides a mechanistic underpinning for the role of NCR1 and NPC2 in trafficking and activity of this drug in cells.

The new addition reads:

"Interestingly, we were able to measure binding of the non-fluorescent ligand, edelfosine, due to its' competitive properties towards NBD-Sph indicated by the change in NBD fluorescence (Figure 4). As mentioned earlier, crystals of NTD and edelfosine are formed for x-ray crystallography, indicating the presence of lipidated NTD. Together these data support binding of edelfosine to the NTD. Previous research shows that Δ NCR1 yeast is resistant to the antifungal drug (41), indicating an interaction between edelfosine and NCR1 in the yeast vacuole. Others have reported that edelfosine targets the plasma membrane and endoplasmic reticulum (ER) by altering the sterol distribution in the membranes (55). Edelfosine might be internalized through endocytosis to the vacuole (41) followed by translocation to the ER. This strongly implies that edelfosine should exit the vacuole by binding to the NTD and passing through the tunnel of NCR1 to obtain its' cytotoxic effect. Additionally, these experiments confirm the binding of a phospholipid analog, adding to the broader spectrum of substrates that can bind to the NTD of NCR1."

Reviewer #2 (Comments to the Authors (Required)):

This manuscript addresses the comparison between the N-terminal domains of mammalian cholesterol transport protein, NPC1, and its yeast homolog, NCR1. The NPC2 proteins of both species are also investigated. The manuscript uses x-ray crystallography and several biophysical methods with sterol and lipid analogs to characterize the similarities and differences of these lipid transporters. A major conclusion is that the yeast NPC2/NCR1 transport system has broader substrate specificity than the human proteins.

I found the overall paper to be interesting and significant, but I found several issues that I think need to be addressed in a revised manuscript.

We thank the reviewer for the positive feedback.

1. There are some serious issues with the assays for binding of lipid derivatives to the N-terminal domain of NCR1 and NPC2.

A. I don't see any value in measurements of binding constants when using concentrations above the critical micelle concentration.

We agree that measuring binding constants when lipids exceed their critical micelle concentration (CMC) is different from measurements below the CMC, but we do not concur with the conclusion, that such experiments are without any value. Many studies have addressed the issue of how self-associating ligands affect binding isotherms, and it has been concluded that binding curves above the ligands CMC can deviate from those below the CMC (e.g. [1,2] in reference list below). Supporting this notion, we have shown previously that the (apparent) dissociation constant of NBD-PC below its CMC is much lower than above, but also, that the shape of the binding curves differs, in line with the aforementioned theoretical work. Future studies are needed to systematically explore the impact of micelle formation on ligand affinities, particularly, because self-associating ligands are a common theme in biochemistry as amply documented for hydrophobic drugs, sterols or phospholipids. In the manuscript, we make it clear in the text that we refer to apparent dissociation constants, and that only values for the exact same conditions should be preferred:

"In these cases, the determined K_D -values should be interpreted as apparent dissociation constants since the measured increase in fluorescence signal depends both on the binding equilibrium and the thermodynamics of micelle-to-monomer interconversion"

and

"Thus, the K_D -values for binding of NBD-PC, NBD-PS and NBD-Cer to the proteins in this study cannot be directly compared to each other or to the K_D -values measured for NBD-PI and NBD-Sph, as for the latter lipids all binding studies were carried out with concentrations below their respective CMC"

In the manuscript, we exclusively compare the dissociation constants measured for lipid binding to the NTD of NCR1 and NPC2 for the same lipid, not between lipids. By comparing the same lipid binding to different proteins at the same concentrations, we expect the lipid to form micelles to the same extent, and the difference between the measured dissociation constants is solely due to the different affinities of the respective proteins.

If we attempted to measure binding of all lipids below their CMCs, we would have trouble detecting fluorescence from the NBD-group for the monomeric form, as for some NBD-lipids, the CMC is simply too low for measuring monomer-protein association reliably. Additionally, in the manuscript, we wanted to compare our results of lipid binding to the NTD to the results of binding to NPC2 from

Moesgaard ([3] in reference list below). Therefore, we used a 1 μ M lipid suspension, as in our previous study.

The results obtained by measuring binding above the CMC of the lipid, indicate that NPC2 might function as a general lipid solubilizer compared to the NTD that most likely depends on the hand-off of single lipid molecule, which we also find interesting, which is why we conclude in lines 310-312 that:

“Based on this strong interaction, it is unlikely that NPC2 will hand over these substrates to the NTD. It can thus be postulated that NPC2 acts as a general lipid solubilizer in the vacuole and might have other interaction partners.”

B. I don't see the value of the edelfosine experiments. It is not shown to bind in the binding pocket, and it could be a non-competitive inhibitor. This seems to be unrelated to the rest of the manuscript, and I would recommend deletion of this section.

We find the edelfosine experiments relevant and valuable, but agree we do not sufficiently explain this in the manuscript. To convince the reader and reviewer of the relevance of these experiments we have added a short paragraph in the Discussion in lines 338-349. The new text emphasizes the role of NCR1 and NPC2 in exporting edelfosine from the vacuole, which could underline the observed edelfosine resistance in yeast lacking the NPC transport system. We feel that our results on edelfosine binding by these proteins provides a mechanistic underpinning for the role of NCR1 and NPC2 in trafficking and activity of this drug in cells.

The new addition reads:

“Interestingly, we were able to measure binding of the non-fluorescent ligand, edelfosine, due to its’ competitive properties towards NBD-Sph indicated by the change in NBD fluorescence (Figure 4). As mentioned earlier, crystals of NTD and edelfosine are formed for x-ray crystallography, indicating the presence of lipidated NTD. Together these data support binding of edelfosine to the NTD. Previous research shows that Δ NCR1 yeast is resistant to the antifungal drug (41), indicating an interaction between edelfosine and NCR1 in the yeast vacuole. Others have reported that edelfosine targets the plasma membrane and endoplasmic reticulum (ER) by altering the sterol distribution in the membranes (55). Edelfosine might be internalized through endocytosis to the vacuole (41) followed by translocation to the ER. This strongly implies that edelfosine should exit the vacuole by binding to the NTD and passing through the tunnel of NCR1 to obtain its’ cytotoxic effect. Additionally, these experiments confirm the binding of a phospholipid analog, adding to the broader spectrum of substrates that can bind to the NTD of NCR1.”

Through the experiments, we detect a competitive binding between edelfosine and NBD-sphingosine. The intensity of prebound NBD-sphingosine decreases after addition of edelfosine (Figure 4A), indicating an exchange of NBD-sphingosine with edelfosine. Additionally, if no lipid occupied the binding pocket of the NTD, crystals were not formed, but crystals were formed with both ceramide and edelfosine, which strongly indicates binding of these. The poor-quality data is most likely due to the mobility of the long acyl chains as mentioned in the discussion:

“The delipidated NTD did not crystallize, but crystals formed when C16 ceramide and edelfosine were added. Unfortunately, the data quality was not sufficient to confidently model these ligands, especially the long, flexible acyl chains that likely protrude from the binding pocket. However, these results prompted us to investigate the ability of the NTD to bind these compounds with biochemical assays.”

2. The structures of the pac-lipids need to be shown.

Pac-lipid structures were added as Figure 5B. As a result, the figure legend expanded by one panel and now reads:

Figure 5. NTD crosslinking with pac-lipids

(A) Schematic illustration of liposome crosslinking assay performed for experiments shown in (C) and quantified in (D). (B) Structures of pac-lipids used in the experiments. (C) Representative gel image with crosslinking profile of purified NTD. Liposomes containing 1.5 mol% of indicated pac-lipids (1 μ M pac-lipid) were incubated with 1.5 μ M NTD and UV-crosslinked. Protein-lipid complexes were subjected to click reaction with AF647-picolyl azide and visualized by SDS-PAGE. In-gel fluorescence of AF647 was normalized to loaded protein in each well based on Coomassie staining. (D) Normalized in-gel fluorescence is quantified as +UV/-UV signal ratio. N = at least 3 independent experiments.

3. The increase in fluorescence upon incubation with the pac-lipids seems extremely small. What does it mean to have a 2-fold increase in fluorescence? What percent of the NTD proteins have been labeled??

The small increase in fluorescence is caused by the frequently occurring high background in the -UV condition, caused probably by non-specific labelling of proteins by AF647 dyes. All data with pac-lipids are shown as a ratio between crosslinked (+UV) and non-crosslinked (-UV) conditions. The background is an issue which we could not completely overcome so far, even with several other azide-conjugated fluorophores that we tested. We cannot calculate the percent of labelled NTD proteins due to this but also because there are no quantitative AF647 fluorescence standards that can be used to estimate the quantity of fluorescence from SDS-PAGE gels (especially with small quantities of proteins). We are interested to generate such standards in the future, to make our assays semi-quantitative. For now, we can only compare the relative crosslinking efficiencies.

4. They show models of different orientations of the NBD group binding to the NTD of NCR1 vs. NPC2. They mention in the Discussion that it might be interesting to check this. I believe it is essential for interpreting these data.

This is a good idea, but it will require a lot of extra work since we would have to perform X-ray crystallography on NBD-lipids binding to the NTD and NPC2. Thus, while certainly interesting, this would be a separate project. Another possibility would be to do MD simulations as done for NPC2 and phospholipids ([3] in reference list below). This, however, would also be very time-consuming and by itself not conclusive for inferring the orientation of the NBD-group in the binding pocket. This is a follow-up project and is beyond the scope of the current manuscript.

Figure S1A needs a more complete figure legend. It is very hard to understand as written.

We agree that the figure legend mentioned by the reviewer can be more detailed and expanded upon the existing figure legend. Changes have been made, not only to Figure S1A, but to the figure legend overall for improved clarity, in the text and indicted in purple below:

Figure S1. Purification and crystallization of the NTD with all N-glycosylation sites

(A) Construct used to over-express containing the NTD that includes the signal peptide (SP), a representative SDS-PAGE gel of an NTD purification, a size-exclusion chromatography trace and

crystals obtained from the concentrated peak fraction at 10.5 mL. ~~its purification by IMAC and SEC as well as representative crystals.~~ Samples for SDS-PAGE were collected throughout the purification. Lane 1: marker, lane 2: IMAC load, lane 3: IMAC flow-through, lane 4: wash flow-through, lane 5: wash + β -cyclodextrin flow-through, lane 6: wash flow-through, lane 7: G20 flow-through, lane 8: G40 flow-through, lane 9: SEC load, lane 10: SEC fraction 2, lane 11: SEC fraction 3, lane 12: SEC fraction 4, lane 13: SEC fraction 6, lane 14: peak SEC fraction 7 and lane 15: SEC fraction 8. **(B)** The NTD contains 16 cystine residues, all of which are involved in forming disulfide bridges. The NTD also has three *N*-glycosylated residues and are partitioned to the opposite face compared to the eight disulfide bridges. ~~Location of eight disulfide bridges and three *N*-glycosylated residues.~~ **(C)** SDS-PAGE of the NTD treated with various deglycosylases enzymes. The size of the NTD decreased when treated with PNGase F, Endo F1 but remained unchanged when treated with Endo F2, Endo F3 and was inconclusive with Endo H because of overlapping size with the NTD – see Materials and Methods for details on cleavage sites. **(D)** Packing of four NTD monomers, Chain A to Chain D, in the asymmetric unit of the crystal. **(E)** Examples of density in grey ~~for surrounding the glycan branches from Chain A to Chain D. chains from each monomer in the asymmetric unit.~~

References

1. L.W. Nichol, G.D. Smith, A.G. Ogston (1969) The effects of isomerization and polymerization on the binding of ligands to acceptor molecules: Implications in metabolic control. *Biochimica et Biophysica Acta (BBA) - General Subjects* 184: 1-10. [https://doi.org/10.1016/0304-4165\(69\)90092-0](https://doi.org/10.1016/0304-4165(69)90092-0).
2. M.J. Sculley, L.W. Nichol, D.J. Winzor (1981) Interactions between micellar ligand systems and acceptors: Forms of binding curves. *Journal of Theoretical Biology* 90: 365-376. [https://doi.org/10.1016/0022-5193\(81\)90317-9](https://doi.org/10.1016/0022-5193(81)90317-9).
3. Moesgaard L, Petersen D, Szomek M, Reinholdt P, Winkler MBL, Frain KM, Muller P, Pedersen BP, Kongsted J, Wustner D (2020) Mechanistic insight into lipid binding to yeast niemann pick type c2 protein. *Biochemistry* 59: 4407-4420. doi:10.1021/acs.biochem.0c00574

Referee Cross-Comments

I agree with Reviewer 1 that this is a valuable contribution.

I would like the authors to take my review into consideration before returning a final version of the manuscript.

As requested, we have done this. Thanks again to both reviewers for the positive and constructive feedback that has improved the manuscript.

October 11, 2024

RE: Life Science Alliance Manuscript #LSA-2024-02990R

Dr. Bjørn Panyella Pedersen
Aarhus University
Department of Molecular Biology and Genetics
Gustav Wieds Vej 10
MBG-AU
Aarhus C, Danmark 8000
Denmark

Dear Dr. Pedersen,

Thank you for submitting your revised manuscript entitled "Structural and biochemical analysis of ligand binding in yeast Niemann-Pick type C1-related protein". We would be happy to publish your paper in Life Science Alliance pending final revisions necessary to meet our formatting guidelines.

- please be sure that the authorship listing and order is correct
- please add ORCID ID for secondary corresponding author-they should have received instructions on how to do so
- please add the Twitter handle of your host institute/organization as well as your own or/and one of the authors in our system
- please consult our manuscript preparation guidelines <https://www.life-science-alliance.org/manuscript-prep> and make sure your manuscript sections are in the correct order

A. FINAL FILES:

B. MANUSCRIPT ORGANIZATION AND FORMATTING:

Sincerely,

October 16, 2024

RE: Life Science Alliance Manuscript #LSA-2024-02990RR

Dr. Bjørn Panyella Pedersen
Aarhus University
Department of Molecular Biology and Genetics
Universitetsbyen 81
MBG-AU
Aarhus C, Danmark 8000
Denmark

Dear Dr. Pedersen,

Thank you for submitting your Research Article entitled "Structural and biochemical analysis of ligand binding in yeast Niemann-Pick type C1-related protein". It is a pleasure to let you know that your manuscript is now accepted for publication in Life Science Alliance. Congratulations on this interesting work.

DISTRIBUTION OF MATERIALS:

Again, congratulations on a very nice paper. I hope you found the review process to be constructive and are pleased with how the manuscript was handled editorially. We look forward to future exciting submissions from your lab.

Sincerely,
